# Security Analysis of the MQTT-SN Protocol for the Internet of Things

José Roldán-Gómez [1,*,†] , Javier Carrillo-Mondéjar [2,†], Juan Manuel Castelo Gómez [2,†]
and Sergio Ruiz-Villafranca [2,†]

1   Department of Computer Science, University of Oviedo, 33003 Oviedo, Spain
2   Institute of Informatics (I3A), University of Castilla-La Mancha, 02071 Albacete, Spain
*   Correspondence: roldangjose@uniovi.es
†   These authors contributed equally to this work.

**Abstract:** The expansion of the Internet of Things (IoT) paradigm has brought with it the challenge of promptly detecting and evaluating attacks against the systems coexisting in it. One of the most recurrent methods used by cybercriminals is to exploit the vulnerabilities found in communication protocols, which can lead to them accessing, altering, and making data inaccessible and even bringing down a device or whole infrastructure. In the case of the IoT, the Message Queuing Telemetry Transport (MQTT) protocol is one of the most-used ones due to its lightness, allowing resource-constrained devices to communicate with each other. Improving its effectiveness, a lighter version of this protocol, namely MQTT for Sensor Networks (MQTT-SN), was especially designed for embedded devices on non-TCP/IP networks. Taking into account the importance of these protocols, together with the significance that security has when it comes to protecting the high-sensitivity data exchanged in IoT networks, this paper presents an exhaustive assessment of the MQTT-SN protocol and describes its shortcomings. In order to do so, seven different highly heterogeneous attacks were designed and tested, evaluating the different security impacts that they can have on a real MQTT-SN network and its performance. Each one of them was compared with a non-attacked implemented reference scenario, which allowed the comparison of an attacked system with that of a system without attacks. Finally, using the knowledge extracted from this evaluation, a threat detector is proposed that can be deployed in an IoT environment and detect previously unmodeled attacks.

**Keywords:** Internet of Things; cybersecurity; protocols; MQTT-SN

## 1. Introduction

The Internet of Things (IoT) is a new technology paradigm that is on the path to change the way we interact with computers and machines. It can be explained as a global network composed of devices (also called *things*) capable of communicating with each other [1,2], which, by doing so, brings with it new possibilities in many fields such as health, the economy, engineering, resource management, and everyday life. As a result, a wide range of industries are researching applications that use this paradigm in the race to having the upper hand in a scenario that has the potential to become a key area in future technology.

This is evidenced by the fast-paced growth rate of the IoT, whose number of devices connected to the Internet surpassed the total of the non-IoT ones [3] in the year 2020; even with the chip shortage caused by the pandemic, there are 12.3 billion IoT devices connected to the Internet, and the predictions are for this figure to grow and reach 27 billion by 2025 [4]. The IoT has several peculiarities. For example, networks are usually dynamic and heterogeneous, giving rise to many different protocols, such as MQTT [5], Bluetooth Low-Energy (BLE) [6], ZigBee (based on the IEEE 802.15.4 standard) [7], and the constrained Application Protocol (CoAP) [8], among others. However, this feature also applies to IoT devices, which can range from the simplest sensors to computing hardware with more resources that sends information to the cloud.

Although this environment has many applications and benefits for industry, from a security point of view, it constitutes a new vector for attacks from cybercriminals [9]. Most of the safety concepts that are widely accepted and applied in network communications have not been taken into account in the IoT. This is due to the fact that IoT systems are usually constrained by a lack of resources related to memories, processors, network bandwidths, and power consumptions, which do not support the implementation of security measures [10].

The diversity of IoT systems and their limitations cause new problems associated with this paradigm [11], for example the need to improve lightweight encryption algorithms and the weak defenses against Denial of Service (DoS) due to the lack of memory, processor power, and bandwidth. Battery consumption gains importance since there are many applications that require the use of devices that only have batteries. Furthermore, it is hard to keep IoT devices updated because they are very heterogeneous [12].

These vulnerabilities are being exploited by criminals to the extent that the evolution of malware specifically targeting IoT Linux has vastly increased in recent years. Figure 1 shows the quarterly growth of samples registered against IoT Linux since the appearance of the Mirai malware, a botnet that interrupted Internet access for millions of people. It can be seen that, from August 2016 to May 2022, there has been a huge increase in samples, fifty-nine-times higher in May 2022 compared to August 2016.

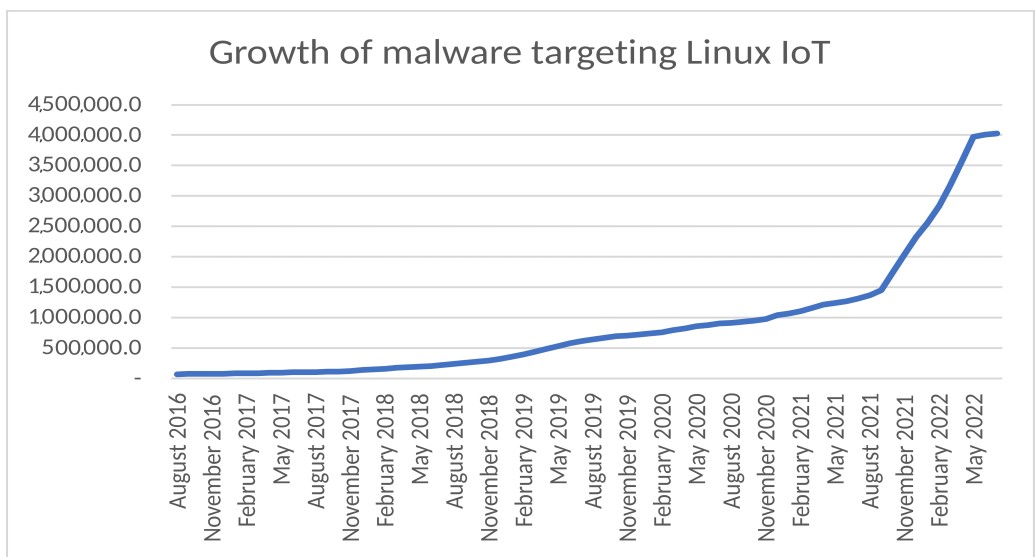

**Figure 1.** Growth of malware targeting IoT Linux.

In the IoT, most attacks are successful because the devices are misconfigured by default, and this indicates that old problems are present in the IoT as well. At this point, it is vital to evaluate the protocols involved in the IoT for a security audit in order to determine their deficiencies and, on that basis, propose improvements.

There are many challenges to be addressed by the research community in the field of IoT cybersecurity. Some of the main ones are listed in the paper *An overview of security and privacy in smart cities IoT communications* by Al-Turjman et al. [13], which identifies the main threats to IoT security. According to the authors, malware, false information, traffic modification, traffic eavesdropping, and identity theft are the most common IoT threats. There are other articles that study the challenges and trends posed by the IoT, and cybersecurity is a key area in all the analyses [14–16], highlighting the protocols and their heterogeneity as a crucial element that needs to be scrutinized. Consequently, this is the field on which we will focus in this work, showing that analyzing protocols from an experimental point of view and extracting their vulnerabilities is critical for IoT cybersecurity research, in addition to assessing the impact of different attacks that exploit the vulnerabilities exposed by Al-Turjman et al. in their work [13].

This research is focused on MQTT-SN [17], which is based on its predecessor, namely MQTT. While MQTT is a lightweight protocol, which uses a publish/subscribe messaging transport scheme, MQTT-SN is the newest version and has been designed to be even lighter than MQTT. The reasoning behind the choosing of this version is due to it being optimal for environments with resource-constrained devices such as the Industrial Internet of Things (IIoT) [18] and yet not having received the focus of detailed research from the security research field. Under these circumstances, this paper aims to firstly find deficiencies in MQTT-SN, secondly to exploit them, and thirdly, to evaluate their impact on the whole IoT scenario. After this evaluation, we recommend an attack detector designed to discover attacks in IoT environments. The advantage of this threat detector is its ability to detect anomalies and unmodeled attacks, which is crucial in combating novel attacks such as those proposed in this work.

This proposal is a substantial extension of the work *Security Assessment of the MQTT-SN Protocol for the Internet of Things* [19]. Schematically, the main contributions of this article are as follows:

- A practical analysis of MQTT-SN protocol vulnerabilities.
- An implementation of attacks based on the vulnerabilities discovered.
- An evaluation of the impact of the different attacks.
- A threat detector, which has already been published, is suggested and tested with the implemented attacks.

This paper is organized as follows. Section 2 describes the MQTT-SN protocol and the main differences between it and MQTT [5]. The state-of-the-art is reviewed in Section 3. A presentation of the MQTT-SN system deployed is made in Section 4. The attacks discovered are described and evaluated in Section 5. Section 6 describes a novel threat detector designed for IoT environments. Finally, we draw our conclusions in Section 7.

## 2. MQTT-SN Protocol

The MQTT-SN [17] protocol is a connectivity protocol based on MQTT [5], which operates at the application layer. MQTT is a topic-based protocol, which makes it possible to create a publish/subscribe-based topology in which each device can subscribe to or publish information about a topic. Thus, there is a broker that manages the topics and the connected devices in the network. This topology is very useful for the IoT, as it allows the transmission of information generated by sensors to central nodes.

MQTT-SN was designed to be lightweight (even more than MQTT [5]) and works with wireless communications. This means that it is characterized by link failures, a short message size, low bandwidth, and low overheads, among other features. MQTT-SN was designed to be similar to MQTT, but there are a few differences. Firstly, MQTT-SN includes *TopicId*, which replaces the topic name in MQTT. *TopicId* is a 16-bit integer, which acts as the topic name, with the *REGISTER* command being able to negotiate the mapping between *TopicId* and the topic name. Secondly, MQTT-SN provides a sleeping client mechanism. This feature enables clients to shut themselves down and save power for a while. MQTT-SN allows us to update or delete the *will* message, which allows devices to notify other clients about an ungracefully disconnected client. Finally, another important aspect is that MQTT-SN works over the User Datagram Protocol (UDP) [20].

UDP, which operates at the transport layer, introduces an 8-byte header. 6LoWPAN, on the other hand, has header compression mechanisms (in the best case, it can use 4-byte headers), and it also has a Maximum Transmission Unit (MTU) of 127 bytes, which is really small compared to other protocols; for example, IPv6 offers an MTU of 12,800 bytes [21]. Payload sizes will be conditioned by the MTU. In our implementation of the different attacks on a 6LoWPAN network, the maximum payload of 6LoWPAN is 38 bytes, and consequently, the UDP payload is 30 bytes. It is important to mention that MQTT-SN operates over UDP, but does not need to operate over 6LoWPAN. The study and implementation of the attacks described in this analysis focused exclusively on MQTT-SN, so the complete study of the protocols of other layers is outside the focus of this work. In Figure 2, which depicts

an MQTT-SN-based network, we can observe how one operates. The publication on a topic is represented with an arrow of the same color as that topic. The grey rectangles symbolize MQTT-SN clients, and the colors inside them represent the topics to which they are subscribed. This scheme allows us to easily understand MQTT and MQTT-SN. The client with *Id* 1 is subscribed to topics 1, 2, and 3; the client whose *Id* is equal to 2 is subscribed to topics 1 and 2; finally, the client whose *Id* is equal to 3 is subscribed to topics 1 and 3. If the client with *Id* number 2 publishes on topic 3, this message is received by clients 1 and 3. The same happens with topic 2 is published by client 2.

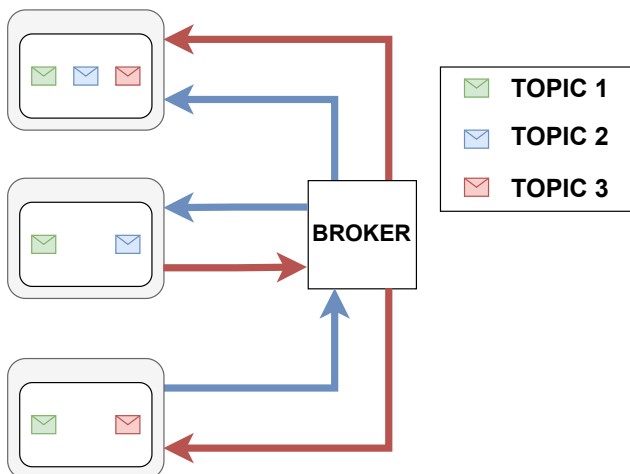

**Figure 2.** MQTT -SN scheme.

### 2.1. Format of the Main Packets

In order to properly understand the attacks to be implemented, a description of the format of an MQTT-SN packet is provided in Table 1, containing the field information common to all packets to be analyzed, with the MQTT-SN packets that are generated or modified in this experiment being detailed below.

**Table 1.** Brief description of the fields common to the different packets.

| Common Fields | Description |
|---|---|
| Length | It determines the length of the packet. |
| MsgType | It defines the type of MQTT-SN packet. |
| Flags | It indicates additional options. |

#### 2.1.1. Connect Message

This type of packet is used to connect MQTT-SN clients to the broker.

Table 2 shows the format of a *Connect*-type packet, the different fields, and the octets they use. The function of these fields is as follows:

- **Flags**: The most relevant ones for a *Connect* packet:
    - *Will*: It defines a message and a topic. In case of an error in the connection between the device and the broker, this message is sent to the chosen topic.
    - *CleanSession*: Enabling this flag causes the client to forget previously subscribed topics. If it is not enabled, the broker computes the topics to which the client was subscribed and keeps them, using the *ClientID* field.
- **ProtocolId**: It determines the version of the protocol being used.
- **Duration**: It contains the value of the keep alive timer of the connection.
- **ClientId**: It uniquely identifies a client connected to the MQTT-SN broker.

**Table 2.** Connect message format.

|  | Length | MsgType | Flags | ProtocolId | Duration | ClientId |
|---|---|---|---|---|---|---|
| Octet number | 0 | 1 | 2 | 3 | 4–5 | 6:n |

### 2.1.2. Publish Message

Publish packets are used when a client sends a message on a topic.

Table 3 shows the format of a *Publish*-type packet, the different fields, and the octets they use. The function of these fields is as follows:

- **Flags**: The most relevant ones for a *Publish* packet:
    - *DUP*: It indicates that the message is a duplicated message, which is sent when no acknowledgment has been received after sending the original one. This behavior occurs with a Quality of Service (QoS) value greater than 0.
    - *QoS*: It determines the QoS level of the Publish message.
    - *Retain*: When a message is published with the *Retain* flag set, the broker saves it as a reference one. Consequently, when a client subscribes to the topic, it automatically receives this message, so it is not necessary for the original sender to publish it again.
    - *TopicIdType*: It indicates the type of identifier found in TopicId. This can be a short name, formed by two characters, or the topic Id formed by 2 bytes.
- **TopicId**: It contains the topic identifier in either of the two formats mentioned above.
- **MsgId**: It uniquely identifies a message when the QoS is greater than zero. It is encoded with 16 bits.
- **Data**: It contains the data of the message to be published.

**Table 3.** Publish message format.

|  | Length | MsgType | Flags | ProtocolId | Duration | ClientId |
|---|---|---|---|---|---|---|
| Octet number | 0 | 1 | 2 | 3–4 | 5–6 | 7:n |

### 2.1.3. Subscribe Message

Subscribe packets are used when a client wants to subscribe to a topic.

Table 4 shows the format of a *Subscribe*-type packet, the different fields, and the octets they use. The function of these fields is as follows:

- **Flags**: The most relevant ones for a *Subscribe* packet:
    - *DUP*: If enabled, the last message with an active DUP fag is received if it exists.
    - *QoS*: It indicates the QoS level required for that topic.
    - *TopicIdType*. It specifies the type of identifier found in TopicId. This can be a short name, topic name, or topic Id.
- **MsgId**: It is used to identify the acknowledgment of receipt, which is sent by means of a *Suback* packet.
- **TopicName or TopicId**: It contains the topic identifier in the format specified in the flag *TopicIdType*.

**Table 4.** Subscribe message format.

|  | Length | MsgType | Flags | MsgId | TopicName or TopicId |
|---|---|---|---|---|---|
| Octet number | 0 | 1 | 2 | 3–4 | (5:n) or (5–6) |

### 2.1.4. Pingreq Message

This type of message is used to know whether a client is connected to the broker. Additionally, in MQTT-SN, it allows taking the device out of sleep mode.

Table 5 shows the format of a *Pingreq* type packet. This format is simpler than those described above:

- **ClientId**: It is an optional field that is used for changing the status of a client from *sleeping* to *awake*. It contains the client identifier.

**Table 5.** Pingreq message format.

|  | Length | MsgType | ClientId (Optional) |
| --- | --- | --- | --- |
| Octet number | 0 | 1 | 2:n |

## 3. State-of-the-Art

As far as the authors of this paper know, there are currently few published studies about the shortcomings of IoT protocols. One of the few is *A survey: Attacks on RPL and 6LoWPAN in IoT* by P.Pongle and G.Chavan [22], which is focused on attacks in the Routing Protocol for Low-Power and Lossy Network (RPL) [23,24]. The RPL protocol enables device routing in Low-power and Lossy Networks (LLNs), which are characterized by the fact that routers usually operate under severe power, memory, and network constraints. This protocol supports point-to-point routing, but also multipoint-to-point and vice versa. It is also highly scalable and can be successfully used with thousands of devices within the LLN. The attacks presented in this paper against RPL typically serve two distinct purposes. On the one hand, we have the attacks that allow a complete or partial denial of service of the network, for example the sinkhole attack. On the other hand, there are also attacks that allow attacking a specific device, e.g., the selective forwarding attack. Finally, there are others that allow the identity of devices to be hijacked to compromise network confidentiality, e.g., the alteration and spoofing attack. The authors also collected certain patterns so that a rule-based Intrusion Detection System (IDS) can be deployed with these patterns.

Another interesting work is [25]. In this paper, the authors conducted a brief study of the Constrained Application Protocol (CoAP), a web protocol designed specifically for devices and networks with limited capabilities. In this paper, the authors used BurpSuite to analyze CoAP request traffic and concluded that the it is not encrypted and also susceptible to attacks.

In [26], the authors describe several security issues with MQTT, but there were no implementations or evaluations of the described attacks. The attacks in this paper, moreover, made an ordinary use of the protocol, not presenting any use of it, which varies from what it is described in the standard. They also contemplated MQTT as a means to create a botnet, although this is not an attack against MQTT; it is the use of it as a means to establish a botnet.

There are several other pieces of research that focus on ZigBee, which is a protocol for low-data-rate and short-range wireless networking. One example is [27], which shows two example attacks: one tries to achieve an eavesdropping attack on a ZigBee network where AES-CCM is used. In order to achieve this, the authors tried to use the same key twice, for example by causing a reset of the nodes. The other attack consists of performing a denial of service, for which they suggested modifying the payload of ZigBee packets, even if there is encryption, to cause an error in the operation of the protocol.

In *Exploiting MQTT-SN for Distributed Reflection Denial-of-Service Attacks* by Sochor et al. [28], the authors suggested to make use of the topology used by MQTT-SN to perform a pro-reflective attack. In other words, they took advantage of the fact that the broker sends messages to all devices subscribed to a topic to perform denials of service against the MQTT-SN network.

A particularly interesting work is *IoT Content Object Security with OSCORE and NDN: A First Experimental Comparison* by Cenk Gundogan et al. [29], which proposes the use of OSCORE, which is a protocol designed to protect end-to-end communications in resource-constrained systems. Its interest lies in them making a comparison with the use of DTLS, which is the encryption protocol usually implemented over UDP. Another interesting proposal that aims to adapt a lightweight encryption method to MQTT and MQTT-SN is the *Lightweight Security Scheme for MQTT/MQTT-SN Protocol* by Ousmane Sadio et al. [30]. To achieve lightweight encryption over the MQTT and MQTT-SN protocol, the authors propose to use ChaCha20, which is a stream cipher protocol. In addition, they also propose to use Poly1305 as a one-time authenticator. In this way, they achieved a lightweight and secure encryption scheme.

Another proposal in the field of lightweight encryption is *Design and evaluation of a novel white-box encryption scheme for resource-constrained IoT devices* by Bang, A.O. and Rao, U.P. [31]. This work focused on generating a scheme capable of protecting IoT environments from white box attacks, those in which the attacker has full view of the execution environment and uses it to break the encryption. The authors, to achieve their goal, propose a scheme to hide the private key in ciphers with elliptic curves.

With regard to comparisons between already existing algorithms, we find the proposal *Safe MQTT-SN: a lightweight secure encrypted communication in IoT* by L.Kao et al. [32], which proposes a secure encryption scheme for MQTT-SN systems. They propose to use the digital signature (ECDSA), hash function, key exchange (ECDHE), ChaCha20, and Poly1305. The authors compared the delay time introduced by the MQTT-TLS handshake with respect to their proposed scheme.

Within the comparisons of the protocols, although not part of the analysis of security, we find a paper that makes a brief comparison from the performance point of view of the IoT protocols, namely *Survey on State of Art IoT Protocols and Applications* by Kumar N.V.R. and Kumar P.M. [33]. It is observed that the results of MQTT-SN are good, especially in the packet loss vs. bandwidth contribution.

A very interesting paper, although not focused on any IoT protocol in particular, is *A Large-Scale Empirical Study on the Vulnerability of Deployed IoT Devices* by Binbin Zhao et al. [34]. In it, the authors surveyed 1,362,906 IoT devices over 10 months. All the conclusions they drew were very interesting, but of special interest is the huge number of vulnerable MQTT servers, a fact that gives an idea of the importance of the matter at hand.

Table 6 shows a brief comparison of the different state-of-the-art works.

**Table 6.** Comparison of the state-of-the-art works.

| Reference | Target Protocol | Highlights |
|:---:|:---:|:---:|
| [22] | RPL | Presents and evaluates different attacks against RPL. |
| [25] | CoAP | Brief analysis of the protocol; concludes that it is not safe due to the lack of encryption. |
| [26] | MQTT | Known attacks against MQTT analyzed. |
| [27] | ZigBee | They describe two different attacks against ZigBee. |
| [28] | MQTT-SN | Reflective attacks employing MQTT-SN to cause DoS. |
| [29–32] | General encryption | All of them propose improvements to achieve a lightweight encryption. |
| [33] | Various | Comparison of protocols; MQTT-SN obtains good results. |
| [34] | Various | Empirical analysis; there are many unsecured MQTT servers. |
| This work | MQTT-SN | Several attacks are implemented and evaluated, and countermeasures are proposed. |

As we can see, the topic of analyzing the security of the MQTT-SN protocol is an unexplored one. Other novelties introduced in this research are the study of the protocol to detect vulnerabilities, as well as how to exploit them. In addition, we analyzed the impact that they have on the system, and we propose a threat detection system to detect such



attacks. Furthermore, as demonstrated by the high number of vulnerable MQTT servers currently deployed, working on the security of the lighter version MQTT-SN is necessary before this protocol is used globally and replaces or coexists with MQTT.

## 4. MQTT-SN Baseline Scenario

The first step in performing a security analysis of the MQTT-SN protocol in order to test attacks and evaluate their impact is to define a legitimate scenario without attacks and evaluate its performance. This scenario is referred to as the baseline. This is important because it makes it possible to compare the impact of each attack (which exploits a shortcoming in the protocol) on the system against the baseline. The evaluation can be focused on features such as packet size, packet rate, etc., and, on that basis, try to determine the impact of the attacks discovered.

The proposed baseline scenario is composed of six devices: one is defined as the broker and the rest as regular devices (which perform the function of a simple sensor to acquire data to be sent to the broker). Within this set of regular devices, the baseline scenario includes one device with certain privileges, which is subscribed to a confidential topic, and four others without privileges. The overall behavior is quite simple: each non-privileged mote is subscribed to a topic referred to as */topic1*. In addition, these non-privileged devices retrieve the information generated by other devices under the */topic1* label. The information generated as */topic1* consists of random floats with no practical meaning. This information could be, in a real scenario, the temperature taken in a room, the humidity in a vineyard, or the vital signs of a patient in a hospital. The generation and handling of this randomly generated information is beyond the scope of this work. In this case, the objective is to evaluate MQTT-SN as an information exchange protocol between the broker and the subscriber nodes.

As mentioned above, the baseline scenario also includes a privileged mote, which generates a random float and publishes it in the other topic referred to as */topic2* in each cycle. The privileged mote generates a symbolic message, which is confidential, and it is useful to evaluate a scenario that manages confidential information. These devices do not contain additional protection measures, but the differentiation of these measures makes it easier to observe the handling of the different topics and their contents. Finally, the broker manages all the connections, processes all the information collected by the devices, and centralizes the network.

Figure 3 shows the baseline scheme, which is composed of devices that generate information on both topics, namely */topic1* and */topic2*, and the broker. In this case, the device with an orange background color, which is located at the bottom left, is subscribed to topic 2, which simulates the sending of critical information. Including this device is useful to evaluate how the system can be altered when using two different topics. By establishing this scenario as the baseline, we can extract valuable information about the regular (not under attack) behavior of the network. Below, we compare the baseline scenario against scenarios with the system under attack.

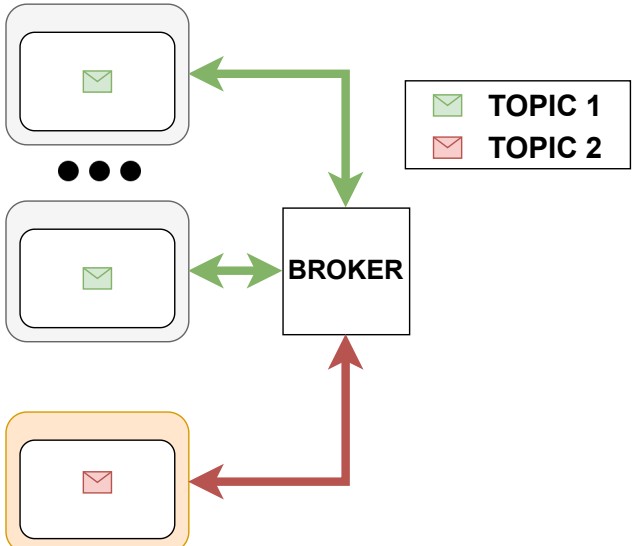

**Figure 3.** MQTT-SN baseline scenario.

## 5. MQTT-SN Attacks and Performance Evaluation

This section enumerates the shortcomings discovered in the MQTT-SN protocol, as mentioned in Section 4, which an attacker can exploit to compromise the information collected between nodes (affecting integrity and confidentiality) or to compromise the entire network and/or devices (affecting availability) in an IoT infrastructure. All the attacks discovered, which represent the main contribution of this paper, are described in the following sub-sections. In addition, these attacks are implemented and their impact is evaluated in this section. For this purpose, the baseline scenario was deployed. Once an attack has been successfully demonstrated, different countermeasures to prevent or mitigate it are presented.

The implementation of these attacks was performed using a network simulator called Cooja, which is Contiki's network simulator, as it allows the experiments to be much more reproducible and parameterizable than a real scenario. Cooja provides an intuitive User Interface (UI), which offers a graphical representation of our network and devices and a toolkit to perform different actions. Contiki [35] is an open-source operating system designed for the IoT, which provides a set of features and tools to design, implement, and deploy IoT applications and systems. It is built on a TCP/IP stack and offers lightweight preemptive scheduling on an event-driven kernel, which is a very motivating feature for the IoT.

This way, the combination of Cooja and Contiki allowed us to deploy a virtualized IoT network and study its behavior. It is widely used to develop applications, but it can be used to design, implement, and study application layer attacks over 6LoWPAN (network layer) on several different devices. In order to capture the traffic of the virtualized network, it was redirected from the broker to the host device, then Wireshark was used to obtain the traffic and analyze it later [36].

### 5.1. Trash-Inject Attacks

The first attack described aims to introduce unexpected elements into the legitimate MQTT-SN application.

Before performing this attack, it is necessary to introduce a malicious device into the network. In order to do so, it is either necessary not to have an authentication mechanism or to gain access to the broker by guessing or having access to the credentials. In our implementation, we assumed that we had access to the network due to a lack of authentication mechanisms, which is something very common in MQTT and MQTT-SN servers [34], so a realistic scenario is proposed.

Once that access to the broker has been gained, the attack consists of introducing malformed information in a specific topic or in a set of topics. The range of possibilities in this attack is very wide since, if the attackers know the logic of the application on which they are going to inject information, they may be able to modify its behavior arbitrarily. However, even without knowing the operation of the application into which malformed information is introduced, this attack can have a very negative impact.

This attack requires prior knowledge of the system's topics, but by default, it is possible to subscribe to all topics using the # character, so knowing all the topics in the system is not a difficult task. Another possible way is to take advantage of the short topic name format of MQTT-SN as it is easy to scan topics when they are encoded with two characters.

Figure 4 shows the format of the *Publish*-type packets. As can be seen, the packet fields that are key to carrying out the attack are marked in red. In this case, the *topic Id* field indicates the topic where the malicious information is to be inserted, and the *data* field contains the information to be entered. The content of these two fields may vary depending on the attacker's target. As mentioned above, if the attacker knows the logic of the system, he/she can inject specific data to cause a specific system failure. However, he/she can also introduce pseudo-random data (trash) to cause a system malfunction.

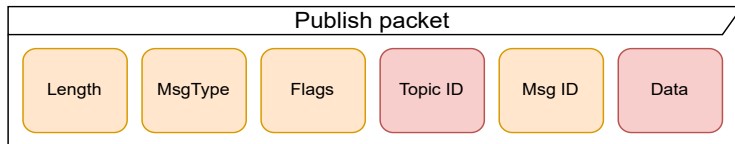

**Figure 4.** Publish packet format.

Figure 5 shows how the attack works, with the attacker having the ability to inject messages into a used topic. The device on the right, which has a red background color, represents the attacking device. As can be observed, in this case, the attacker injects trash into topic 1 and the broker distributes that trash to all clients subscribed to that topic. It should be noted that this attack can be carried out by contaminating several topics simultaneously in a similar way. In our implementation, we only focused on one topic, because it is easier to quantify the impact in this way.

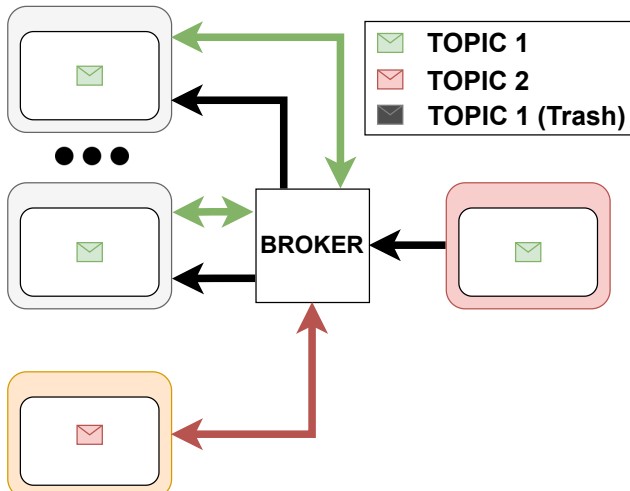

**Figure 5.** MQTT-SN trash injection scheme.

Evaluation of the Trash Injection Attack

Before diving into the evaluation, it is necessary to clarify that the baseline system works by publishing random numbers to simulate a sensor. Thus, our application expects a numerical value, and if it receives a different kind of data, it will not work properly.

Figure 6a shows the expected topics (numbers) versus the unexpected values (character strings). This type of graph is recurrent throughout the paper, each point representing a packet, so that we can easily observe the behavior of the system. As we can see, the malicious device easily sends many malicious packets. This ability to inject traffic can lead to a malfunctioning of the MQTT-SN application. In this experiment, the client application receives text when it expects numeric data, what causes the application to stop working. However, in other applications, the consequences can be more negative, since an attacker is capable of injecting inject information into an application.

Figure 6b shows the amount of accumulated bytes received by the broker over time, proving that this attack can be performed without significantly increasing the traffic.

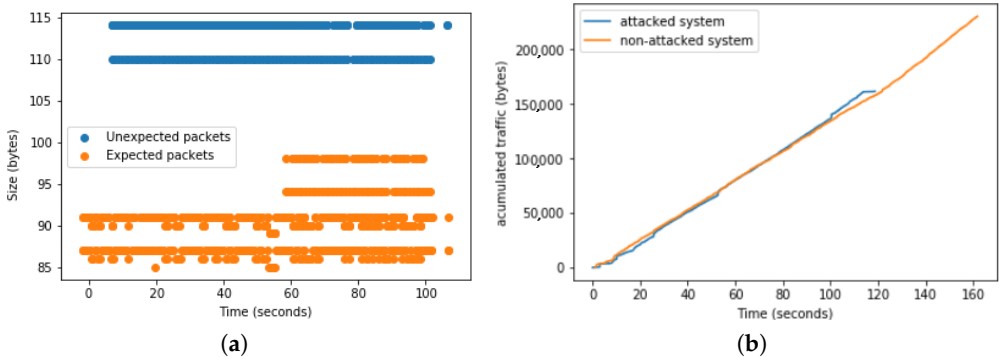

**Figure 6.** Trash injection attack evaluation diagrams. (**a**) Expected against unexpected packets. (**b**) Cumulative traffic of baseline scenario vs. trash injection attack scenario.

This kind of attack can be partially stopped if we enable the user/password request in our MQTT-SN network. The problem is that this solution will not work if the attacker infects a mote that has been previously authenticated. Another relevant aspect is that this attack is even easier to use against MQTT-SN than against MQTT, because the use of UDP allows an IP spoofing attack to be carried out easily.

### 5.2. Information Leaks

Another security weakness of MQTT-SN is that (by default) it might be possible to subscribe to all the topics, which, in many cases, can have serious consequences. The intrinsic problem is that the topics in MQTT-SN are defined as a structured hierarchy. For example, we could subscribe to or publish on the topic referred to as *house/bedroom2/sensor1*. This structure is perfectly valid for a house with several bedrooms and a few sensors inside each room. In this example, we can subscribe to all the topics from all the bedrooms by simply subscribing to *house/#*. The # operator will obtain a subscription to all the rooms in the house. Furthermore, if we subscribe to the /# topic, we will subscribe to all the topics in the broker. It is useful to understand that when the # character is used, it is not mapped to a numeric *Id*. In addition, when a client subscribes to a topic, you can use the name, the *Id* of the topic, or the short name of the topic, which can lead to obtaining sensitive information by using the # character. It could be possible to prevent subscription to /#, as this receives a simple solution in MQTT in which there are no short topics, with each topic being linked to a number. However, if the attack is against an MQTT-SN network, the attackers could use the short name format to iterate over possible topics easily. They could also use the topic *Id*, since, as it is encoded with two octets, its iteration is feasible. Once the attacker manages to subscribe to all the topics, they obtain access to all the messages in the network, even if we have sensitive information in them. This implementation is not a classic eavesdropping attack in which a Man in the Middle (MitM) scheme is sought as a general rule. In this case, it is necessary to subscribe to the topics, with the vulnerability consisting of the possibility of subscribing to unknown topics.

As we can see in Figure 7, packets of type *Subscribe* are used, and the attackers only need to manipulate the *topic Id* field to subscribe to the topic of their choice.

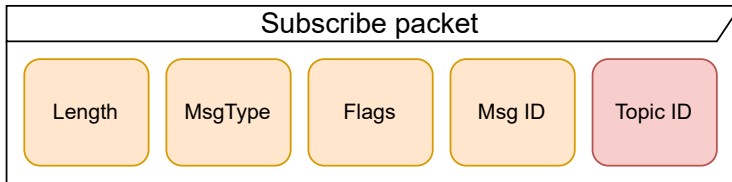

**Figure 7.** Subscribe packet format.

Figure 8 shows the scheme of the attack. As we can see, the attacking device tries to subscribe to all topics by iterating the numeric *Id*. In this way, the malicious device is able to extract information from all the topics in the system.

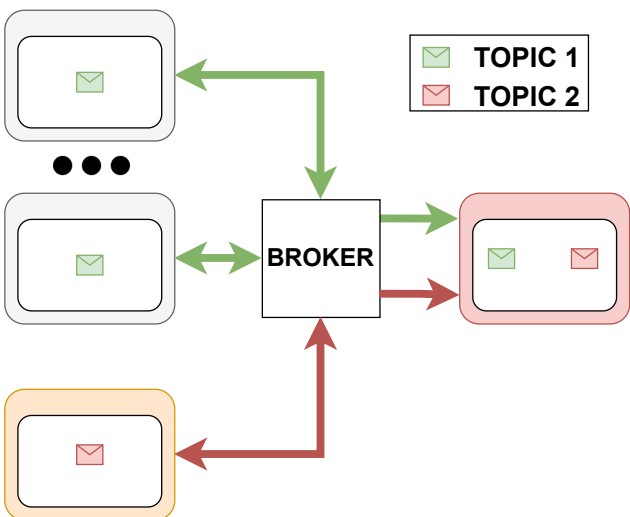

**Figure 8.** MQTT-SN information leak scheme.

Evaluation of Information Leak Attack

This attack can be dangerous if the system is misconfigured. In this case, an attacker can subscribe to any topic in the system and retrieve all the information generated in the IoT network. Figure 9 shows how easily an attacker can access confidential information. This protocol is highly vulnerable to this type of attack because the publish/subscribe scheme means that the attacker does not need a previous man in the middle attack to carry it out.

```
▸ Internet Protocol Version 6, Src: aaaa::1, Dst: aaaa::c30c:0:0:8
▸ User Datagram Protocol, Src Port: 1884, Dst Port: 1884
▾ MQ Telemetry Transport Protocol for Sensor Networks
  ▾ Message
      Message Type: Publish Message (0x0c)
      Message Length: 20
      0... .... = DUP: No
      .00. .... = QoS: Fire and Forget (0x0)
      ...0 .... = Retain: No
      .... ..00 = Topic ID Type: Normal ID (0x0)
      Topic ID: 2
      Message ID: 0
      Message: confidential
```

**Figure 9.** Malicious mote subscribed to confidential topic.

Although this attack does not modify the behavior or performance of the network, it greatly impacts the confidentiality of the system. This can be avoided by making use of MQTT-SN's built-in authentication.

*5.3. Disconnect Wave*

Disconnect wave is a DoS attack that exploits a big failing in the MQTT-SN protocol. The specification of MQTT-SN indicates that each client has a unique *Id*, so if a new client tries to register this *Id* again, the latter gains the *Id* and ejects the previously created connection. Once we know the *Id*, the attack can be carried out immediately. The *Id* field has a variable length between 1 and 23 characters, and it is sent by the client when making the connection. This means that if a bad security policy is used by the client application, it is possible to guess the *Id*. In our application, this *Id* is a device counter to ensure that it is unique.

Figure 10 shows the format of the *connect* packets used to steal the *Id* of legitimate clients and removethem from the system. In this case, the *client Id* must be modified depending on the targeted client.

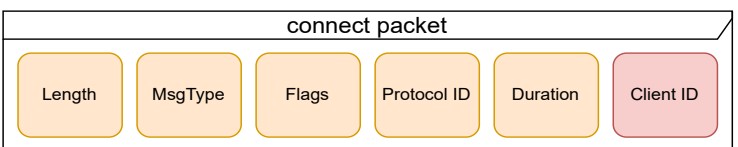

**Figure 10.** MQTT-SN connect packet format.

As can be seen in Figure 11a, the malicious device connects to the *Ids* of legitimate devices to eject them. To achieve this, it iterates over the *Ids* of the legitimate devices. If guessing this *Id* is not trivial, it is possible to generate an MitM scheme to know the identifiers of the legitimate devices. Figure 11b shows the result of the attack, and as we can see, the legitimate devices are expelled.

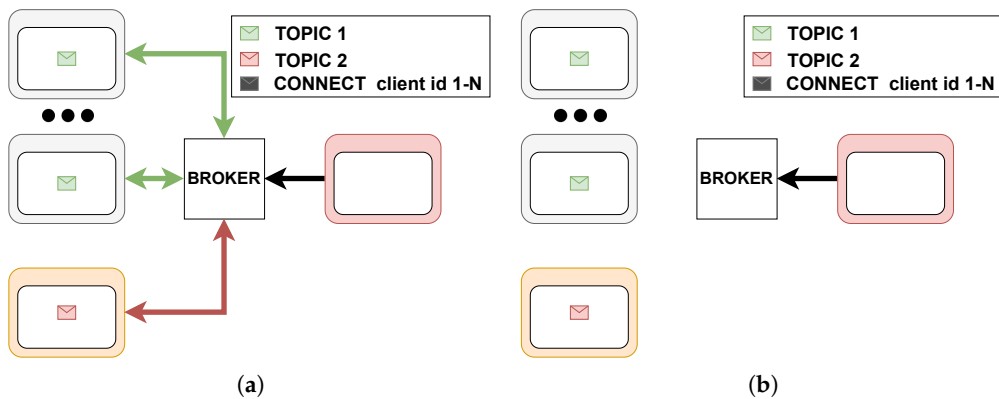

(**a**)                                                                                     (**b**)

**Figure 11.** MQTT-SN disconnect wave scheme. (**a**) MQTT-SN disconnect wave, implementation of the attack. (**b**) MQTT-SN disconnect wave, result of the attack.

Evaluation of the Disconnect Wave Attack

As is shown in Figure 12a, the broker receives a flood of connect commands, but it does not send any publish packets. The attacker sends connect packets iterating over the *Id* to expel legitimate devices whose *Id* matches those of these packets. Therefore, we have many *connect ack* packets and zero *publish message* packets, which can ultimately completely override the network.

Even if the attacker performs this attack successfully, as we can see in Figure 12b, legitimate devices keep sending messages. This attack can be stopped with a rules-based IDS, which detects that the network is flooded with connect packets. This attack is an innovation introduced in this work, so it can be difficult to find rules if you want to use a rule-based threat detector, as is the case with other novel attacks due to the lack of knowledge about them.

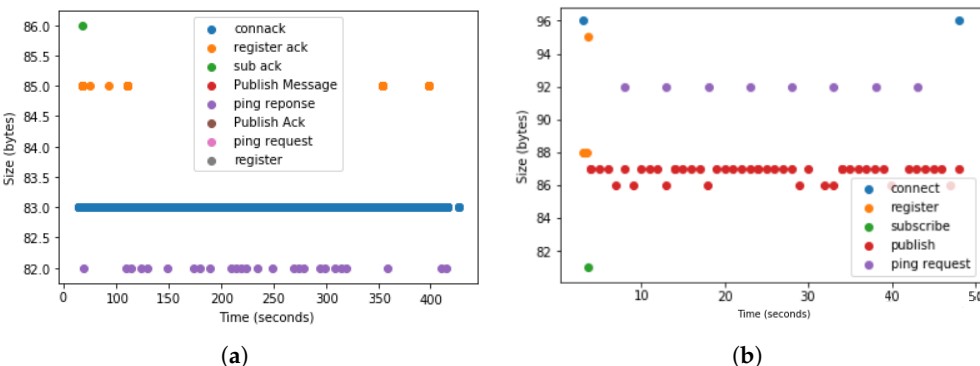

(a)          (b)

**Figure 12.** Disconnect wave attack evaluation diagrams. (**a**) MQTT-SN broker after disconnect wave attack. (**b**) MQTT-SN legitimate client after the disconnect wave attack.

### 5.4. Sniffing Attack

This attack is based on a well-known MitM scheme. It can be performed using layers below the application layer, where MQTT-SN operates. This case is different from the rest of the attacks discussed in this paper because the attackers need to achieve an MitM scheme, a process that requires no use of MQTT-SN features. This attack takes advantage of the unencrypted connection (by default) in MQTT-SN communications, but has the limitation (compared with the others presented in this paper) that it needs a third protocol to complete the MitM scheme. Normally, MQTT-SN is supported with other lower layers such as RPL [24] or 6LoWPAN [21]. For this attack, it is firstly necessary to compromise the communications by attacking these protocols, and once the attacker is in the middle of the communications, he/she has access to MQTT-SN information. The impact of this attack could be extensive: first of all, it can affect confidentiality; secondly, integrity suffers because the attacker can modify the messages; finally, availability is not guaranteed because the attacker can break the communication. Figure 13 depicts an MitM attack over the MQTT-SN protocol. We can see that the malicious device is placed between the broker and the legitimate devices.

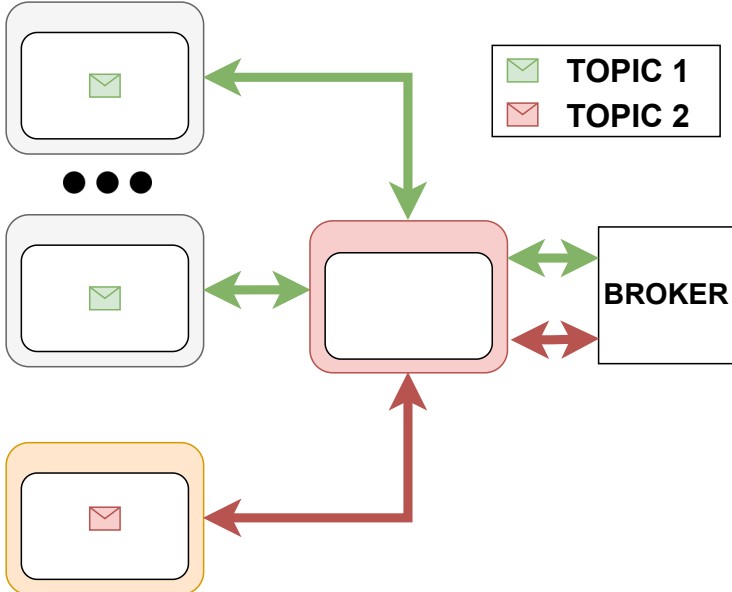

**Figure 13.** MQTT-MitM scheme.

Evaluation of the Sniffing Attack

This attack is successful when there is unencrypted communication, and it must be preceded by a man in the middle attack. In our experiment, a rank attack against RPL [37] was used to achieve the MitM scheme. The consequence of this attack is that the attacker gains access to the information exchanged between the broker and the devices (see Figure 14). This is not a pure IoT attack, but in this context, it can be more dangerous, because when there are constrained devices, it is more difficult to implement an encryption algorithm to protect the communication. One way to tackle this problem is to use Secure MQTT-SN (SMQTT-SN) [38], but this implies an overhead that is introduced in the communication and more resources employed on the devices. This attack does not affect network performance, but then again, it is an attack that compromises system confidentiality.

```
▶ Internet Protocol Version 6, Src: aaaa::c30c:0:0:4, Dst: aaaa::1
▶ User Datagram Protocol, Src Port: 1884, Dst Port: 1884
▼ MQ Telemetry Transport Protocol for Sensor Networks
  ▼ Message
      Message Type: Publish Message (0x0c)
      Message Length: 13
      0... .... = DUP: No
      .00. .... = QoS: Fire and Forget (0x0)
      ...0 .... = Retain: No
      .... ..00 = Topic ID Type: Normal ID (0x0)
      Topic ID: 1
      Message ID: 0
      Message: 14194
```

**Figure 14.** Sniffed pcap view.

### 5.5. Spoofing Connection Via Id

This attack has a more complex and circumstantial implementation than the previous ones. Its objective is to modify the topics to which a client is subscribed, which requires that the legitimate client application does not use the *CleanSession* flag each time it connects to a client.

As seen in other attacks, the client identifier is exploited to expel a client if it is connected when the message is sent, or simply to spoof in case that it is not connected. It is important to remark that, in this scenario, the *CleanSession* flag is enabled. This means that the broker deletes the subscriptions to the different topics of the client with that specific *Id*. The attackers can then subscribe the client to the topics they consider, or simply leave it without any subscription. This completely modifies the reception of information from one or more clients. Figure 15a shows a spoofing attack over MQTT-SN.

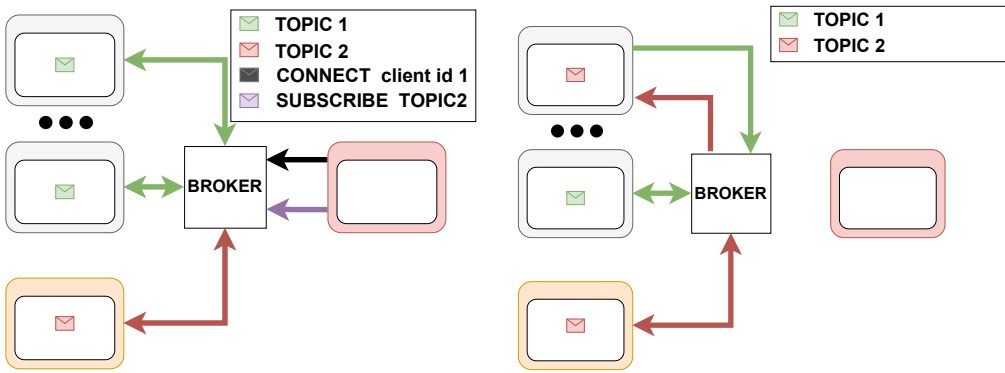

(**a**) Spoofing connection, implementation of the attack.  (**b**) Spoofing connection, result of the attack.

**Figure 15.** MQTT-SN spoofing connection scheme.

In addition, Figure 16 shows the format of the connect-type packet, when the connection is to be spoofed. In this case, in addition to modifying the client *Id*, the attackers have to send the *CleanSession* flag active.

Finally, Figure 15b shows the result of the successful attack. The target *Id* is now subscribed to the topics that the attacker wishes. The malicious device does not need to interact with the broker again.

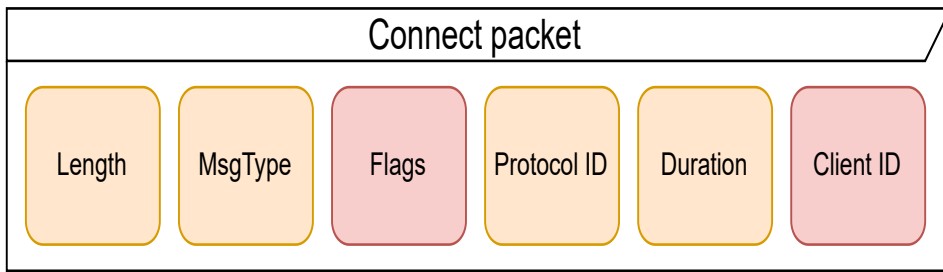

**Figure 16.** Connect packet format (spoofing connection).

### 5.6. Evaluation of the Spoofing Legitimate Clients Attack

This attack is implemented in Cooja by adding a malicious client that expels the target client and overwrites the subscribed topics. Figure 17a shows the normal topic setup for each legitimate device in the baseline scenario, while Figure 17b shows a malicious topic setup for each legitimate device in an attacked system. As can be seen in Figure 17b, an attacker could change the configuration of each device. We can see that, through this attack, we can change the topics to which each legitimate device subscribes. This attack could be avoided by including an IDS in the network, as this can detect when a new device is not authorized to join the topology.

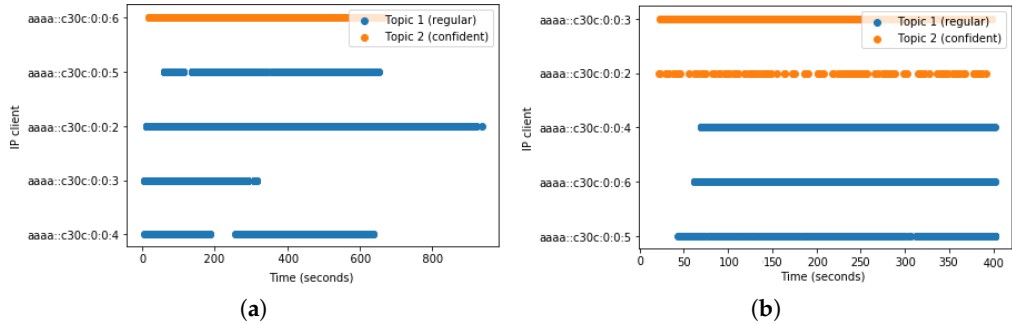

**Figure 17.** Spoofing connection attack evaluation diagrams. (**a**) MQTT-SN topics sent for each device in the baseline scenario. (**b**) MQTT-SN topics sent for each device after spoofing connections.

### 5.7. Congestion Attack

This attack is very common in many protocols, but there is a new factor to take into account when it is implemented over MQTT/MQTT-SN. As the broker must distribute the published messages, an attacker could publish massive messages on a common topic. This means the impact is multiplied by the number of devices that have been subscribed to this topic. There are several approaches to performing the attack. For example, it can create big packets (with 6LoWPAN, there is a maximum of 127 bytes for the maximum transmission unit), or it can send many packets as quickly as possible.

Figure 18 shows the scheme for performing the attack, illustrating how the flow of *topic1* increases with respect to the normal scenario. In this experiment, this was performed by sending messages without delay with the maximum allowed size.

We can see the format of the *Publish*-type packet (Figure 19) in the congestion attack. The only difference with respect to a trash injection attack is that, here, as a general rule, the packet will be sent with the largest possible size, so the *length* field gains relevance,

although it is actually calculated on the basis of the *data* field, which can be used to identify this attack quickly. The size may vary depending on the network protocols used.

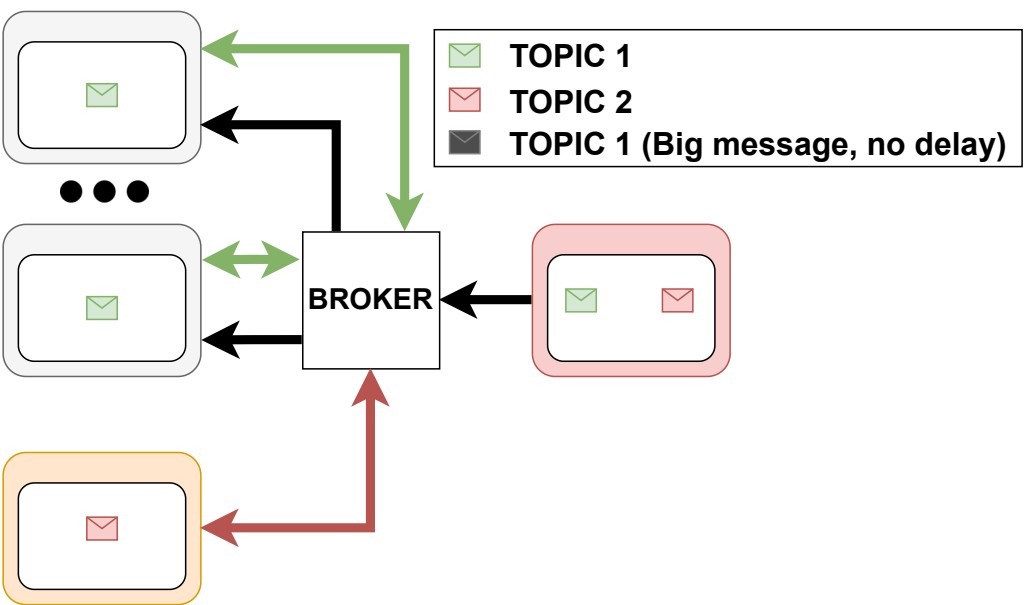

**Figure 18.** MQTT-SN congested accumulated traffic scheme.

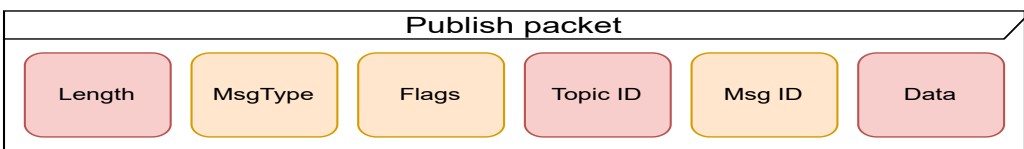

**Figure 19.** Publish packet format (congestion attack).

Evaluation of the Congestion Attack

To demonstrate the impact of this attack, firstly, Figure 20a shows the packets that have been sent by the broker in the baseline scenario, and then, Figure 20b shows the traffic on a compromised IoT baseline system. It can be seen how the broker seems to send fewer messages.

In order to observe the impact more clearly, Figure 20c zooms in on the behavior of the system when it is not under attack, while Figure 20d zooms in on the behavior of the system when it is under attack. These images show an increase in the number of packets received by the broker when the system is under attack.

In this implementation, the attacker floods a topic with two devices that send messages uninterruptedly, and the broker's buffer seems to have overflowed because it stops sending other topics. In this sense, the attacker is easily able to modify the normal behavior of the system. Figure 21 shows the accumulated traffic volume difference between the baseline system and a congested system (flooding a topic with two devices). As we can observe, the attackers considerably increased the traffic of the overall system. This could be a serious issue in an IoT context, because in a small network, this increase will have a negative impact on battery consumption. A network-based IDS could identify congestion in the network and alert the administrator to a possible attack.

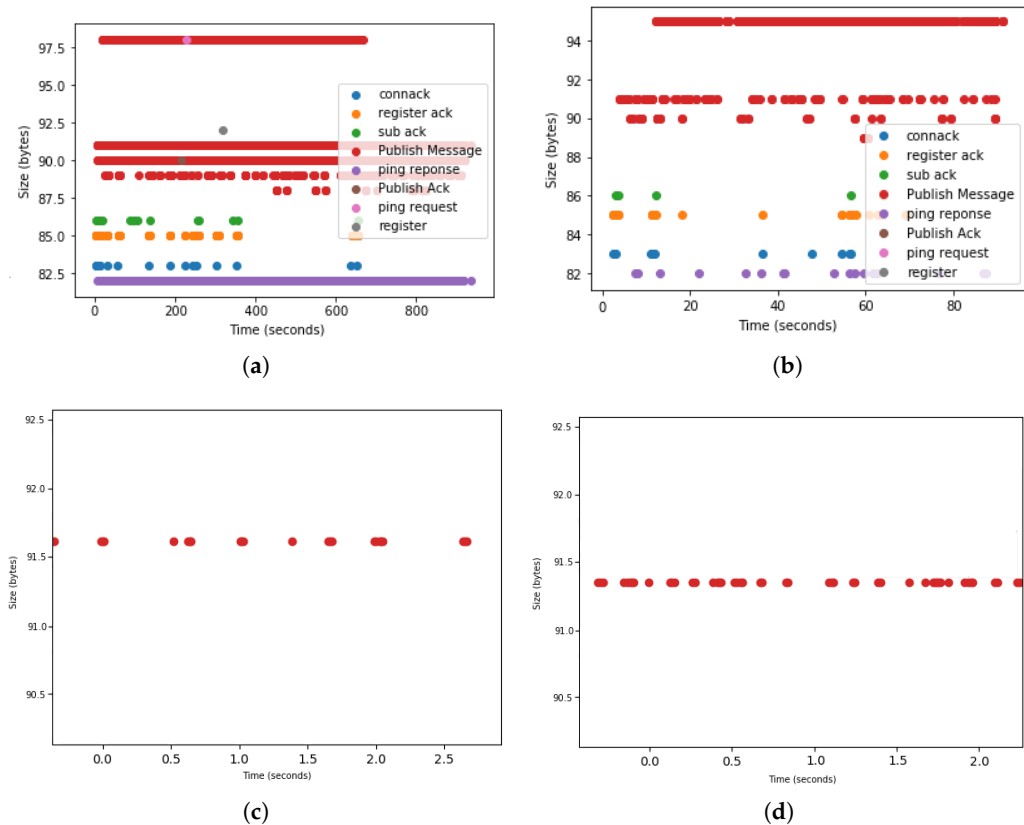

**Figure 20.** Congestion attack evaluation diagrams. (**a**) MQTT-SN traffic in the legitimate scenario. (**b**) MQTT-SN traffic in the attacked system. (**c**) MQTT-SN most-common topic frequency without attacks. (**d**) MQTT-SN most-common topic frequency on an attacked system.

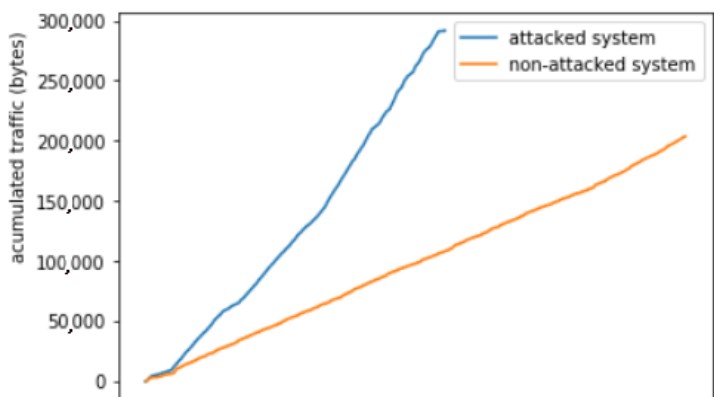

**Figure 21.** Accumulated traffic with topic flooding.

### 5.8. Wake up Wave Attack

The sleeping mode of MQTT-SN allows clients to remain in a sleeping state in order not to waste battery power. While in this state, they do not receive the published messages of the topics to which they are subscribed. This state is reached by sending a *disconnect* message in which a time of duration is indicated. If this time is not specified, the broker simply disconnects the client. When the client is ready to receive messages again, a *Pingreq* message is sent, indicating the client identifier. This sends the client to a state called *awake*, where it can receive messages again. A diagram of how this mechanism works can be seen in Figure 22.

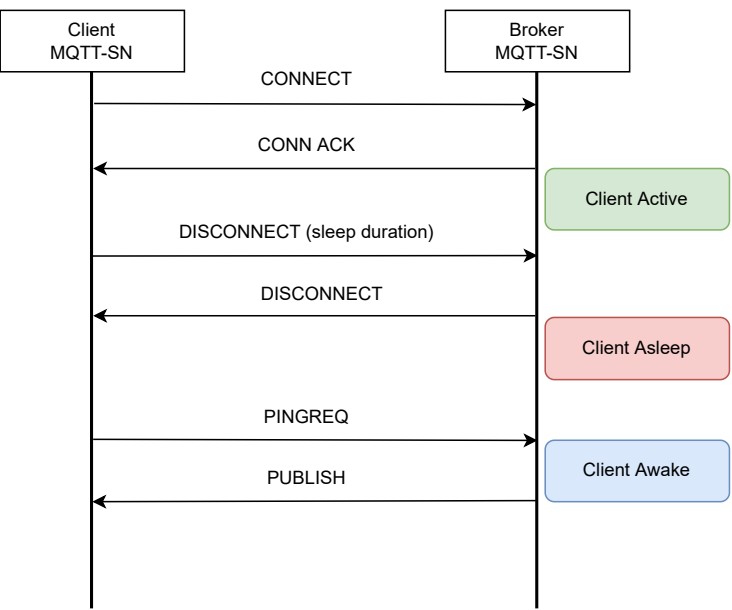

**Figure 22.** MQTT-SN sleeping mechanism diagram.

This attack, which exploits the mechanism introduced in MQTT-SN, aims to prevent clients from sleeping, so that the broker does not stop sending them the various publishes. This can be harmful, especially for devices that rely on batteries to operate. It also increases network traffic.

Figure 23 shows the format of the Pingreq packet. In this case, an identifier of the client we want to wake up is sent. In our implementation, we iterated over the different clients permanently with the aim of preventing any of them from going into sleep mode. A diagram of the attack can be seen in Figure 24.

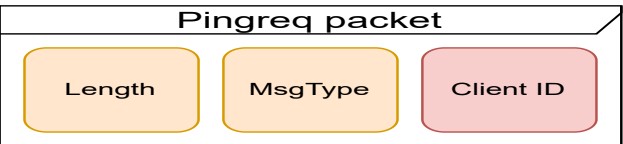

**Figure 23.** Pingreq packet format.

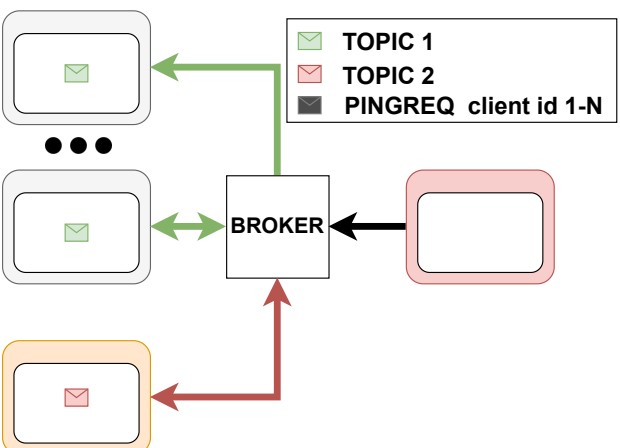

**Figure 24.** MQTT-SN wake up wave attack scheme.

Evaluation of the Wake up Wave Attack

As mentioned above, in this implementation of the attack, it iterates over the different clients indefinitely, preventing any of them from sleeping. This causes, as can be seen in

Figure 25, a huge increase in the system's network usage. The same occurs with the battery usage if the devices are dependent on it, which is usually the case when using sleep mode. To defend against this attack, a rule-based IDS could be interesting since detecting that a single IP address sends multiples Pingreqs with different Ids could be the signature to identify this particular attack.

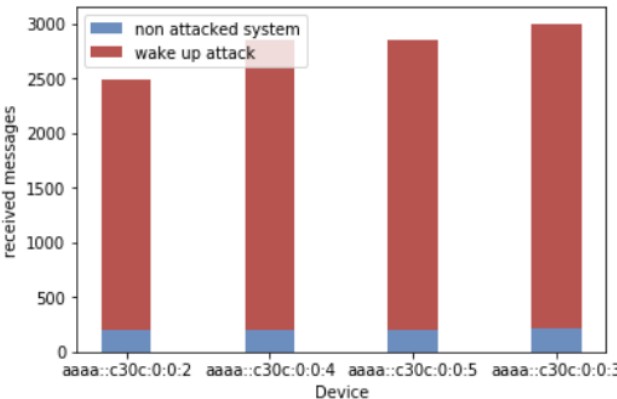

**Figure 25.** Received messages with attack versus those without attack.

## 6. Threat Detector Proposed

This section describes the deployment of the threat detector designed for IoT environments.

As described above, the impact of these attacks when successful is considerable, and deploying typical detection systems can be complicated due to the scarce resources that IoT devices possess. Therefore, we propose a threat detector that is specifically designed to operate in IoT environments [39] and is based on a Complex Event Processing (CEP) engine. CEP is a technology capable of processing and correlating an enormous amount of data. The CEP engine receives simple events, which in this case are network packets. When these simple events meet the requirements set in the CEP rules, a complex event is triggered, which defines a situation of interest, which in this case, is a particular type of attack. CEP is used as the base engine because it has been demonstrated that it is possible to deploy this type of engine in IoT environments with few resources [40–42].

As a general rule, it is necessary for a domain expert to define these CEP rules. However, an architecture capable of generating CEP rules automatically has been designed [39]. This architecture is ideal for detecting the attacks in this paper, as they are novel attacks. This means that, in a real environment, the domain experts cannot define rules for these attacks because they do not know them.

This architecture is based on the use of Principal Component Analysis (PCA) [43] and the Euclidean distance weighted with the explained variance ratio of each component of the PCA model. Each component condenses the information of different features of the events by means of linear relationships between them, and the components are linearly independent of each other. The explained variance ratio allows us to know the weight of each component. By doing so, we can weight each component more accurately when generating the rules. This allows us to reduce the dimensionality of simple events while characterizing the different families of attacks and to improve the performance of CEP rules in the network, computation, and memory domains. In addition, it also allows us to generate anomaly-detecting rules, which are ideal for detecting unknown attacks.

$$f(x) = \begin{cases} 1 \text{ if } f(x) = \sum_i^n \sqrt{(x_i - m_i)^2} \cdot rv_i \leq (\sum_i^n std_i \cdot rv_i) + \alpha \\ 0 \text{ if } f(x) = \sum_i^n \sqrt{(x_i - m_i)^2} \cdot rv_i > (\sum_i^n std_i \cdot rv_i) + \alpha \end{cases} \tag{1}$$

Equation (1) shows the definition of a threshold for a CEP rule generated by our proposal. Simple events, which are network packets in this case, are reduced with the PCA model. Once the reduced simple event $x$ is available, it is sent to the CEP engine. The CEP engine calculates the difference of each component $x_i$ of that reduced simple event with

the mean of that component for each family $m_i$ and weights it with the explained variance ratio of each component $rv_i$. The $\alpha$ element makes it possible to add a bias for attacks with elements that are very far from the mean. If the sum of these weighted differences is less than the sum of the standard deviations $std_i$ weighted with the explained variance ratio of each component, it means that this element is part of that family. Otherwise, it is not part of that family. This allows the generation of rules that detect anomalies by using a rule that detects legitimate system behavior and makes it possible to generate anomaly detection rules when a single event does not correspond to any type of attack and is not part of the system's legitimate behavior. It is important to understand that the real novelty of this threat detector does not lie in Equation (1), which defines the detection threshold. The real novelty is the use of PCA to reduce single events, which reduces the computational load of the CEP engine. This is why this approach is novel and ideal in IoT environments.

Table 7 shows the results of the proposed threat detector against the attacks that were presented in this paper. It should be noted that it was used to detect attacks that are more difficult to detect using rule-based threat detectors. In this case, a single rule was generated that tries to highlight all traffic that is malicious.

**Table 7.** Suggested threat detector results by packet type.

| Packet Type | TP | TN | FP | FN |
|---|---|---|---|---|
| Normal traffic | 0 | 653 | 24 | 0 |
| Disconnect wave | 2357 | 0 | 0 | 0 |
| Trash injection | 76 | 0 | 0 | 0 |
| Congestion attack | 824 | 0 | 0 | 0 |
| Fake Id | 8 | 0 | 0 | 0 |

We can see how the different types of traffic are detected by the detector (Table 8); in this case, it was analyzed packet by packet. The classifier achieves a True Positive (TP) when a packet, which belongs to an attack, is classified as an attack, a True Negative (TN) when a packet, which belongs to normal traffic, is classified as such, a False Positive (FP) when a packet, which belongs to normal traffic, is classified as an attack, and finally, a False Negative (FN) when a packet that belongs to the attack traffic is classified as normal traffic. To accurately assess the effectiveness of the classifier, the following metrics are used:

- Precision $= \frac{TP}{TP+FP}$
- Recall $= \frac{TP}{TP+FN}$
- F1 Score $= 2 \cdot \frac{Precision \cdot Recall}{Precision+Recall}$

**Table 8.** Overall results of the classifier.

| Precision | Recall | F1 Score |
|---|---|---|
| 0.9927 | 1 | 0.9963 |

All metrics are between 0 and 1, with 1 being a perfect score. A high precision metric score indicates that the classifier does not generate many false positives. A high recall score indicates that the classifier does not generate many false negatives, and the F1 score metric combines the two previous metrics to obtain the combined performance.

As we can see, this configuration causes some normal packets to be detected as attacks, but in contrast, allows all attacks to be detected. Prioritizing the recall metric over the precision metric is very common in the cybersecurity field.

In addition, the threat detector we suggest has been tested in other contexts to detect attacks in the IoT environment with very positive results [39], being able to detect attacks against the MQTT protocol with very good computational performance figures.

## 7. Conclusions

The growth of the IoT has meant a drastic change in the technological world. When a new paradigm arrives in a field, it brings many changes with it. In the case of the IoT, having new devices and systems with specific features and requirements has consequently meant the development of new communication protocols. Although it may not be the first element to pay attention to when it comes to discussing IoT security, their importance must not be underestimated, as they are the means to transport information. Data are exchanged in large quantities in the IoT, with some of them being extremely sensitive. For this reason, the research community has shown interest in analyzing some protocols to determine their security level and how they can be exploited. However, the MQTT protocol, while being one of the most-used ones in this environment, has not been studied in detail. The same goes for its lighter version, MQTT-SN, for which, although it has not been yet as successful as MQTT due to its novelty, it is reasonable to think that it may reach similar usage figures.

Under these circumstances, this paper provided an overview of the shortcomings of the MQTT-SN protocol, which is appropriate for IoT scenarios. By compromising MQTT-SN communications, the entire IoT infrastructure can be affected in terms of integrity, confidentiality, and availability. MQTT-SN has inherited security weaknesses from MQTT, its predecessor, which are related to authentication and encryption. In addition, this paper highlighted the fact that MQTT-SN includes new features, such as the sleeping device feature or the short name topic, which make it easier to attack MQTT-SN than MQTT. The performance evaluation demonstrated that security is not an intrinsic feature of MQTT-SN, and it is necessary to investigate, in a new specification, the way to solve security issues while keeping in mind that the devices are resource-constrained and thus limited to executing low-complexity algorithms, as well as most of the protocols are designed to be light and do not support excessive overheads. Finally, this paper showed the fragility of this protocol, meaning it is relatively simple to attack, and as we can see in the previous section, these attacks have a huge impact on the system. Therefore, we proposed a CEP-based IDS capable of operating in IoT environments and detecting unmodeled attacks. This threat detector was tested at MQTT with extremely good results.

The main conclusions and novelties drawn from this work are as follows:

- A practical analysis of the MQTT-SN protocol from a security point of view was carried out, noting that it has several weaknesses that can be exploited.
- Different attacks that exploit the vulnerabilities of the protocol were proposed, and their operation was explained.
- The attacks were implemented, and the impact they have on an MQTT-SN network was measured, allowing the impact of the attacks to be analyzed. This impact is quite large in some attacks, while other attacks have a more circumstantial impact.
- Countermeasures were proposed to mitigate the effect of these attacks. In addition, the use of a threat detector was suggested, which obtained good results, with an F1 score of 0.9963.

**Author Contributions:** Conceptualization: J.R.-G., J.C.-M. and J.M.C.G.; methodology: J.R.-G.; software: J.R.-G.; investigation: J.R.-G., J.C.-M. and J.M.C.G.; data curation: J.R.-G.; draft preparation: J.R.-G.; review and editing: J.M.C.G. and S.R.-V.; visualization: J.R.-G., J.C.-M. and S.R.-V.; supervision: J.M.C.G. All authors have read and agreed to the published version of the manuscript.

**Funding:** This work was supported by the Spanish Ministry of Science, Innovation and Universities and the European Union FEDER Funds (Grant Numbers FPU 17/02007 and FPU 17/03105),by the Spanish Ministry of Economic Affairs and Digital Transformation under the project RTI2018-098156-B-C52, by the Spanish Ministry of Science and Innovation under the project PID2021-123627OB-C52, by the University of Castilla La Mancha (Grant Numbers DO20184364 and PI001482), and by the JCCM (Grant Number SBPLY/21/180501/000195).

**Institutional Review Board Statement:** Not applicable.

**Informed Consent Statement:** Not applicable.

**Data Availability Statement:** The data presented in this study are available upon request from the corresponding author.

**Conflicts of Interest:** The authors declare no conflict of interest.

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
