# Peer review of "Security Analysis of the MQTT-SN Protocol for the Internet of Things"

_applsci, doi:10.3390/app122110991_

Round 1

Reviewer 1 Report (New Reviewer)

In this paper, the authors present security analysis of MQTT-SN protocol that is gaining audience in IoT contexts.

Although the paper provide some analysis, some points need to be addressed before the paper is ready for publication.

1. Section 3 leaves out a lot of related works. The Section is also expected to provide deeper analysis and summaries of previous works.

2. The description of the operation of the protocol in Section 2 is too detailed. A simpler summary of the protocol operation would suffice.

3. The proposed threat detector description is not adequately described. There needs to be a proof of concept or proper imperical evidence to prove that the proposed solution is better than solutions presented in previous works.

Author Response

Dear Reviewer 1.

We thank you for your comments, which we believe have helped to substantially improve the content and presentation of the paper. In the revised version that we are submitting, we have followed all the remarks and recommendations provided in your report. In this cover letter, a detailed list of actions and responses follows, which outline each change that has been made in relation to the specific points raised.

"Section 3 leaves out a lot of related works. The Section is also expected to provide deeper analysis and summaries of previous works".

The explanations and analysis of the works exposed in the related work (Section 3) have been expanded. This allows a better understanding of the articles appearing in this section. In addition, a table, namely Table 6, has been added for a quick comparison of the different works presented in the state of the art.

"The description of the operation of the protocol in Section 2 is too detailed. A simpler summary of the protocol operation would suffice."

A table, namely Table 1, has been created containing the fields that are common to all the packets being analyzed. This makes it possible to simplify the section that analyzes the packet formats.

"The proposed threat detector description is not adequately described. There needs to be a proof of concept or proper imperical evidence to prove that the proposed solution is better than solutions presented in previous works."

Although the main focus of the work is not the threat detector, it has been deployed to detect the attacks we propose in this paper. The results we obtained are very good considering that it has been designed to work in low resource environments.

Along with these changes, the introduction has also been greatly improved, providing a better motivation of the problem and a justification of this work. In addition, the novelties of this work, as well as the conclusions extracted from it, have been explained in a more schematic way to improve readability.

Reviewer 2 Report (New Reviewer)

The goal of this paper, as exposed by the authors, is to propose an attack detector designed to discover attack in IoT environments. Based on experimental evaluation of the wake up wave attack the advantage of this proposal is its ability to detect anomalies and unmodeled attacks, which is crucial in combating novel attacks such as those proposed in this work.

The introduction is too short and does not present in detail the problems, current IoT limitations and challenges of researchers regarding the requirements and cybersecurity scientific results.

Section 3 should contain an in-depth review of state of the art, but presents after MQTT-SN protocol some works, mainly lacking a comparison of the results obtained by researchers.

Throughout the work, abbreviations sometimes appear without explaining them, which denotes a diminished attention of the author. Eg UDP (lines 83 page 2). Regarding UDP, IP, nothing is specified about the OSI model and real-time issues. Specify which Sniffer was used to perform the capture in figure 8. What is the size of the header and payload for IP, UDP and MQTT?

Section 5 it is very well presented providing the expected experimental material and data and the innovation factor brought by the authors.

Equation 1 that defines a CEP rule threshold from section 6 is not sufficiently correlated with the experimental results presented in section 5. There are mainly missing aspects related to threat detector proposed, considering an article submitted to the Applied Sciences - MDPI journal.

It is not clear whether the authors' contributions to THIS publication (apart from what is specified in line 264). The reference section is good, citing new and relevant articles in the research area.

Author Response

Dear Reviewer 2.

We thank you for your comments, which we believe have helped to substantially improve the content and presentation of the paper. In the revised version that we are submitting, we have followed all the remarks and recommendations provided in your report. In this cover letter, a detailed list of actions and responses follows, which outline each change that has been made in relation to the specific points raised.

" The introduction is too short and does not present in detail the problems, current IoT limitations and challenges of researchers regarding the requirements and cybersecurity scientific results ".

The introduction has been improved and expanded to better justify the irruption of the IoT paradigm, the increase in attacks against this paradigm, and the limitations and challenges of this paradigm within the field of security.

"The description of the operation of the protocol in Section 2 is too detailed. A simpler summary of the protocol operation would suffice.

A table has been created with the fields common to the different packages being analyzed. This allows us to reduce the section briefly and avoid redundant information.

"The proposed threat detector description is not adequately described. There needs to be a proof of concept or proper imperical evidence to prove that the proposed solution is better than solutions presented in previous works."

Although the main focus of the work is not on the threat detector, it has been deployed to detect the attacks we propose in this paper. The results we obtained are very good, as we expected. All this with a detector that has been designed to work in low resource environments.

Section 3 should contain an in-depth review of state of the art, but presents after MQTT-SN protocol some works, mainly lacking a comparison of the results obtained by researchers.

The explanations provided for the different state-of-the-art attacks have been expanded. A table, namely Table 6, has also been added to allow a quick comparison of the proposals of each paper. However, it has not been possible to perform a more exhaustive comparison with other works because there is no other evaluation of attacks on MQTT-SN like the one we present.

Throughout the work, abbreviations sometimes appear without explaining them, which denotes a diminished attention of the author. Eg UDP (lines 83 page 2). Regarding UDP, IP, nothing is specified about the OSI model and real-time issues. Specify which Sniffer was used to perform the capture in figure 8. What is the size of the header and payload for IP, UDP and MQTT

All these details have been fixed. The abbreviations have been explained, the layers to which the protocol being introduced belongs are also explained., the size of the headers and payloads of the protocols we obtain in our implementation is also specified, and the sniffer we use, which is Wireshark, is also mentioned.

Equation 1 that defines a CEP rule threshold from section 6 is not sufficiently correlated with the experimental results presented in section 5. There are mainly missing aspects related to threat detector proposed, considering an article submitted to the Applied Sciences - MDPI journal.

Although the main focus of the work is not the threat detector, the detector has been deployed to detect the attacks we propose. We can observe that the results obtained are very good.

“It is not clear whether the authors' contributions to THIS publication (apart from what is specified in line 264). The reference section is good, citing new and relevant articles in the research area.”

The contributions of the paper are now better highlighted, this is done schematically in the introduction. In addition, the conclusions are also schematized to improve the clarity of the contributions obtained.

Round 2

Reviewer 2 Report (New Reviewer)

The authors have addressed most of my concerns satisfactorily. Therefore, I propose the paper for publication.

This manuscript is a resubmission of an earlier submission. The following is a list of the peer review reports and author responses from that submission.

Round 1

Reviewer 1 Report

DoS usually stands for denial of service. Nevertheless, “denegation” seems also to be used in the literature. Please, add an explanation if these two are both the same concept.

Line 93: missing citation

Section 4 is quite verbose and repetitive, and, hence, unclear when read at first. Maybe, I simple graphical representation of the scenario would help. Additionally, a rationale for the scenario should be provided. In particular, what is the role of the privileged mote? And how the confidentiality of the messages is provided as it seems that MQTT-SN does not ensure any?

Line 172: “is” is unnecessary.

As for the description of particular attack vectors and their evaluation it is quite confusing for the reader who is interested in one particular attack that he or she must jump back and forth throughout the paper in order to understand it as a whole. My suggestion would, thus, be to put the description of the attack and its evaluation together. Hence, also the repetition between the two may be eliminated. Nevertheless, I understand the current separation, as my suggestion would require to restructure the paper.

Many descriptions of the attacks are inadequate or vague. More details of the implementation should be provided. For example, trash-inject attack seems to consist of a pre-phase for discovering topic ids. I think think this requires a separate paragraph of text. The authors state (line 183) to describe to ways to implement the attack. However, the following text is not an implementation for me: these is just differentiation on attacking a topic or many topics. How is the implementation of many topic attack different from attacking one topic? Are there any implementation differences or is the latter only a generalization of the former?

Another example of vague description is in Section 5.2. First, it seems that topics in MQTT-SN are addressed by numbers and in MQTT by strings. How is then operator # applicable to MQTT-SN? Is this section focused on MQTT or MQTT-SN? How big can an interval for topics ids be (line 206)? Where is the information leak here? How is this type of attack connected to the “classic eavesdropping” mention in the previous section?

Section 5.3: How many client ids are possible? Is is feasible to make such an attack if ids are unknown?

Is the sniffing attack really the attack on MQTT-SN or is it the attack on the lower layers?

References to Figure X should have capital initial letter (line 253, line 265).

A topology was defined (line 284): How? What are virtualized IoT networks and scenarios that are used in the evaluation?

As already mentioned, there is some repetitive text in every subsection of the evaluation as the attacks were already described above.

Figures are often referenced, but with insufficient explanation what the figure represents. Moreover, captions below the figures are minimal. For example, Figure 18, what exactly is the idea to show here? Plot axes are unexplained. Why is the time scale from 880 s to 980 s? What is the number +1.561545e9 below the plot? What does “accumulated traffic” means? Furthermore, sending anomalous packets is impressing. Please explain how is this connected to injection of traffic? How this influences the system? Are such packets filtered out or not? Do they damage the system? And how can this lead to the remote execution of code? How can a code be injected by sending a string instead of integer in the packer? Do you have a particular vulnerability in the OS in mind?

Figure 19 and also many others are inadequately explained.

What is the goal of Section 7? Where are the results (line 452?)

I could continue with similar comments as above for the rest of the evaluation section. In my opinion, a scientific paper should provide a clear and detailed enough explanation for every technique, statement, result etc.

Nevertheless, I find the paper topic interesting and worthy of research. I hope my review to be taken as a constructive comments in order to improve it.

Reviewer 2 Report

Presented article of authors titled "Security analysis of the MQTT-SN protocol for the Internet of Things" is a contribution in the field of the security of IoT environment.

The article consists of six chapters. It contains 21 pages of text, including the list of 33 references, 31 figures, 1 table and 1 mathematic formula. The abstract is reasonably extensive and sufficiently explanatory. The list of references is fully sufficient.

In the first chapter, authors provide introduction into the issue. Second chapter deals with MQTT-SN protocol. Third chapter shows list of related works and papers in the field. The fourth chapter introduces baseline scenario. Fifth chapter enumerates attacks, that authors used on IoT infrastructure. Sixth chapter extensively presents the results of attacks shown in previous chapter. Seventh chapter proposes a CEP-based IDS capable of operating in IoT environments. Last chapter – conclusion sums up the paper.

Regarding of the paper, to the best of my knowledge, I think that the topic of is actual and proposed solution should work. Moreover, this attack overview can be helpful not only to researchers, but also to professional public.

In my opinion, the chosen language is at very good level, the paper is understandable. All objectives are clearly defined, and the steps during experiments are documented in detail. Considering mentioned I recommend the article for publication in the MDPI Applied Sciences journal.

Reviewer 3 Report

The authors illustrate some attacks in the MQTT-SN protocol and propose a threat detector for the protocol. The topic has scientific and practical value. However, there are some issues in this manuscript.

1. Most of the contents of the manuscript are examples rather than security analysis in the real sense, which is inconsistent with the title of the paper. The instances of attack cannot substitute for formal security analysis.

2. The attack detection method proposed by Equation 1 is too simple and less innovative.

3. The Figures in the text are not clear enough.

To sum up, the reviewer cannot recommend the acceptance of this manuscript.

Round 2

Reviewer 1 Report

The authors have restructured and improved the paper as requested. However, the main problem of vague descriptions and unclear experiments remains in the paper.

Reviewer 3 Report

1. There is too little analysis in the manuscript and most of it is just illustrative. Even without formal protocol analysis, there can be formal expression of various attacks.

2. The description of threat detector is not complete, clear or innovative.